# Evaluation of Thirdhand Smoke Exposure after Short Visits to Public Facilities (Noraebang and Internet Cafés): A Prospective Cohort Study

**DOI:** 10.3390/toxics10060307

**Published:** 2022-06-07

**Authors:** Myung-Bae Park, Boram Sim

**Affiliations:** 1Department of Gerontology Health and Welfare, Pai Chai University, Daejeon 35345, Korea; parkmb@pcu.ac.kr; 2HIRA Research Institute, Health Insurance Review and Assessment Service (HIRA), Wonju 26465, Korea

**Keywords:** cotinine, passive smoking, public health, public facilities, thirdhand smoke

## Abstract

We aimed to evaluate the degree of thirdhand smoke (THS) caused by short-term exposure to smoking-related substances. To this end, we evaluated the change in concentration of a smoking-related urine biomarker in volunteers before and after visiting public spaces where there is likely THS exposure. We hypothesized that a visit to such public spaces would result in an increase in such biomarkers. Participants visited one of the predetermined facilities (noraebang, PC café) and revisited the same facility after 24 h, spending around 2 h per visit. We selected creatinine-corrected urine cotinine (CUC) as a biomarker to evaluate THS. In addition, we collected nicotine-derived nitrosamine ketone (NNK) from surface dust at each site with cotton swabs (diameter of 2.5 cm). We examined whether CUC concentration significantly changed across three time points (baseline, first visit, and second visit) via repeated-measures analysis of variance (RM-ANOVA). Moreover, we analyzed the interaction to determine whether cigarette smell affects the CUC concentration. Finally, CUC and dust NNK were analyzed with Pearson’s correlation. The CUC concentration did not increase from baseline to the first visit, but increased from the baseline to the second visit (Diff = Ln [0.565] ng/mg, P < 0.01). Further, the CUC concentration increased from the first to the second visit (Diff = Ln [0.393] ng/mg, *p* < 0.01). In the case of the interaction effect, there were statistically significant differences in CUC concentration depending on the smell of smoke in the facility (Diff = Ln [0.325], F value = 4.438, *p* value = 0.041). The change in CUC concentration from baseline to the second visit (r = 0.562, *p* < 0.001) and from the first to the second visit (r = 0.544, *p* < 0.001) were correlated with NNK concentration. We evaluated whether a short stay in a facility with smoke-related substances that adhere to the surrounding environment would expose individuals to THS even if they do not smell or are directly exposed to cigarette smoke. We confirmed that even two relatively short stays (approximately 2 h each) in a facility in which people had previously smoked can lead to THS exposure.

## 1. Introduction

Smoking, whilst being the leading single cause of morbidity and mortality, with a greater mortality rate than the sum of the effects of all other causes (e.g., drinking, traffic accidents, and acquired immune deficiency syndrome), is a preventable factor [1]. Passive smoking is as harmful as direct smoking, and was designated a cause of lung cancer in 1992 by the US Environmental Protection Agency [2]. Approximately one million people worldwide are estimated to die from passive smoking every year [3]. Passive smoking in young children and adolescents induce premature death, sudden infant death syndrome, lower respiratory tract infection, asthma, and pneumonia, among other conditions, and passive smoking in adults also has several adverse effects, including lung cancer and cardiovascular disease [1,4].

For these reasons, the World Health Organization enforced the Framework Convention on Tobacco Control (FCTC)—the first international treaty in public health—in 2005. As of 2022, 182 countries worldwide are bound by this treaty, covering 90% of the global population [5]. Article 8 of the FCTC consists of guidelines on the protection from exposure to tobacco smoke. Consequently, countries are continuing to expand their smoke-free zones. As a result of such endeavors, the rate of secondhand smoke (SHS) exposure has been declining worldwide [6,7].

Despite these efforts, however, thirdhand smoke (THS) has recently surfaced as a newly recognized threat. SHS refers to passive smoking in which a nonsmoker is exposed to mainstream or sidestream smoke due to smoking by another person. THS, on the other hand, refers to the exposure to smoke-related substances, as opposed to the direct tobacco smoke, by being present in a place in which someone has smoked or by coming into contact with a smoker. In other words, the toxic substances and microparticles in tobacco smoke adhere to a smoker’s hair and clothes, as well as to living spaces such as walls, curtains, and furniture, and these can contaminate smokers and nonsmokers alike [8,9].

THS poses a health risk by altering and destroying the DNA structure [10], increasing the incidence of cancer in nonsmokers [11], and emitting the same toxic substances as SHS [12]. Cotinine and nicotine concentrations reportedly increase among individuals who spend time in places in which smoking was likely to have occurred, such as places of accommodation, rental cars, and used cars [13,14,15,16]. Nonsmokers living with a smoker (even if SHS does not occur within the house) reportedly exhibit a high urine cotinine concentration [17]. Despite the globally declining trend of SHS, evidence on THS is limited. In a study in which Korean national data were used, Sim [18] reported that the SHS exposure rate among nonsmokers dropped by more than 80% since 2010 but that their cotinine concentrations did not considerably decrease. One possible reason for this phenomenon is THS.

Research on THS is still in its initial stages. Published studies to date have only considered exposure of more than a day, such as the scenarios of living with a smoker, staying in a hotel, or renting a car. We are not aware of published studies of short-term exposure to THS. Therefore, in this study, we aimed to evaluate the degree of THS caused by short-term exposure to smoking-related substances. To this end, we evaluated the change in concentration of a smoking-related urine biomarker in volunteers before and after visiting public spaces where THS exposure is likely. We hypothesized that a visit to such public spaces would result in an increase in such biomarkers.

## 2. Methods

### 2.1. Setting and Participants

This prospective cohort study was conducted in Korea, and enrolled current nonsmokers. A nonsmoker is defined as a person who has not smoked a cigarette for two years and has not used electronic cigarettes, chewing tobacco, etc., in any form. We chose internet cafés and noraebang (Korean style karaoke) as the settings of exposure for this study, both of which are public places with a risk of THS exposure [19]. On the day of the study, each participant visited one of the predetermined facilities and revisited the same facility after 24 h, spending around 2 h per visit. Moreover, to claim that short-term visits to target facilities cause THS exposure, possible SHS and THS exposure from other places were minimized. First, we recruited 45 participants who do not live with a smoker to prevent SHS and THS exposure at home. Second, participants were instructed in advance not to visit a noraebang, internet café, or pub where SHS and THS may occur and to avoid coming into contact with people who were smoking three days prior to the study. Lastly, when participants selected facilities, they were instructed to exclude facilities in which people were currently smoking or in which cigarette smoke was observed. 

Additionally, participants had the freedom to choose the place they visit (noraebang or internet café), and this was not dictated by the research team. Accordingly, the research team made adjustments to avoid participants crowding into only one type of facility. Furthermore, more than two participants in one place did not visit repeatedly. Our sample was of medium effect size (Cohen’s F = 0.25), significance level (α) = 0.05, and power (1-β) = 0.95 when calculated by G-Power based on a two-tailed test for repeated-measures analysis of variance (RM-ANOVA).

### 2.2. Outcome Measures 

#### 2.2.1. Biomarkers

Cotinine is the most representative substance for the measurement of smoking exposure. It has a half-life of 18–20 h and is used to evaluate smoking exposure for up to 7 days [20]. It is usually collected from urine, blood, or saliva. However, creatinine-corrected cotinine is more sensitive for the evaluation of smoking substances extracted during passive smoking. Hence, we selected creatinine-corrected urine cotinine (CUC) as the biomarker to evaluate THS [21]. Urine was self-collected by the participants 24 h after the visit, and the research team subsequently stored it at −70 °C. Cotinine was measured using liquid chromatography-tandem mass spectrometry on an API 4000 and a TurboIonSpray interface (multiple reaction monitoring mode; Applied Biosystems/MDS Sciex, Waltham, MA, USA), and creatinine concentrations were measured using a Toshiba 200FR analyzer (Canon Medical Systems Corporation, Ōtawara, Japan).

#### 2.2.2. Environmental Indicators

The evaluation of THS necessitates the detection of tobacco-derived environmental substances that may affect smoking exposure at the sites visited. We selected nicotine-derived nitrosamine ketone (NNK) as the environmental indicator. NNK is the most representative tobacco-specific N-nitrosamine produced by the reaction of nicotine with nitrous acid (HONO) in the air [22]. We collected NNK from the surface dust at each site with cotton swabs (diameter of 2.5 cm) upon the first visit of participants to the sites. For noraebang, samples were collected by the participants near the computer and sound equipment inside the room or from the top of a dusty door. For internet cafés, participants collected samples from the desks or computers used by the public. The collected dust was sampled on a filter paper by adding 100 mM ammonium acetate. Samples were analyzed using an Agilent 1260 Fast Resolution Liquid Chromatography system (Agilent Technologies, Inc., Santa Clara, CA, USA) combined with a triple quadrupole mass spectrometer equipped with a TurboIonSpray TM source (AB SCIEX Pte. Ltd., Singapore City, Singapore).

#### 2.2.3. Smoke Smell

The most intuitive way to confirm exposure to smoking is by smell. In our study, the participants were asked to respond to the question “Did you smell cigarette smoke upon entering the facility?” using a five-point Likert scale. The responses “very strongly,” “strongly,” and “moderately” were considered to indicate having smelled cigarette smoke, while the responses “rarely” and “not at all” were considered to indicate not having smelled cigarette smoke. The responses (smelling or not smelling cigarette smoke upon entering the facility) were consistent between the first and second visits. Since the rates of the first and second visits for cigarette smell were the same, we did not distinguish between the first and second visits in our analysis.

### 2.3. Statistical Analysis

We examined whether CUC concentration significantly changed across three time points (baseline, first visit, and second visit) via RM-ANOVA, followed by Fisher’s least significant difference (LSD) post-hoc test for post-hoc comparisons. In addition, we analyzed the interaction to determine whether cigarette smell affects the CUC concentration. Finally, we determined whether the changes in CUC concentration were due to smoke-related substances in the facilities visited by analyzing the correlation between CUC and dust NNK with Pearson’s correlation coefficient. All statistical analyses, with the exception of descriptive statistics, were converted to natural logs for analysis.

## 3. Results

### 3.1. Chracteristics of Participants and Target Venues

A total of 45 participants (21 men and 24 women) were enrolled. Twenty-three (51.1%) participants visited an internet café, while 22 (48.9%) visited a noraebang. Among the participants, 40 (88.9%) stated that there were no windows for ventilation in the facilities they visited, and 34 (75.6%) stated that they did not smell cigarette smoke upon entering the facility. The mean duration of their stay was 120.8 min during the first visit and 122.1 min during the second visit (Table 1). 

### 3.2. Dust NNK and CUC Concentrations at the Baseline and First and Second Visits

The geometric mean (GM) dust NNK concentration was 114.5 pg/mg (median: 105.0 pg/mg). The GM CUC concentration increased across the time points, from 47.9 ng/mg (median: 46.3 ng/mg) at the baseline to 56.2 ng/mg (median: 56.5 ng/mg) at the first visit and 80.7 ng/mg (median: 79.8 ng/mg) at the second visit (Figure 1).

### 3.3. Results of RM-ANOVA

RM-ANOVA revealed that the model satisfied the assumption of sphericity (Mauchly’s W = 0.933, *p* = 0.233, Greenhouse-Geisser = 0.937). The CUC concentration differed among the time points (F = 6.593, *p* < 0.01). The CUC concentration did not increase from baseline to the first visit, but increased from the baseline to the second visit (Diff = Ln [0.565] ng/mg, *p* < 0.01). Furthermore, the CUC concentration also increased from the first to the second visit (Diff = Ln [0.393] ng/mg, *p* < 0.01) (Figure 1).

Additionally, we analyzed the interaction effect to examine whether exposure to THS differs between facilities that smelled of smoke and those that did not; there were statistically significant differences in CUC concentration depending on the smell of smoke in the facility, the concentration increased further in the absence of tobacco smell (Diff = Ln [0.325], F value = 4.438, *p* value = 0.041) (Figure 2).

### 3.4. Correlation Analysis

The change in CUC concentration from baseline to the first visit was not correlated with NNK concentration (r = 0.144, *p* < 0.001). However, the change in CUC concentration from baseline to the second visit (r = 0.562, *p* < 0.001) and from the first to the second visit (r = 0.544, *p* < 0.001) were correlated with NNK concentration. Finally, NNK concentration was not correlated with the smell of smoke in the facility (r = 0.093, *p* =0.544) (Table 2).

## 4. Discussion

To the best of our knowledge, this is the first study in which the detection of smoke-related substances in the body was evaluated after short stays in public places that were anticipated to result in exposure to THS. CUC concentrations were measured at baseline and after each of two 2 h visits to an internet café or noraebang. Participants’ CUC concentrations were statistically significantly increased after the second visit from the baseline and from the first visit. According to our results, two visits to such public spaces are unsafe in terms of THS. Larger studies are needed to determine whether a single visit should also be deemed harmful. It is possible that the participants were exposed to smoke-related substances in places other than the target facilities. However, dust NNK concentration in the target facilities were associated with participants’ CUC concentrations. Therefore, NNKs at the study facilities were the most likely cause of elevated CUC concentrations, which supports our hypothesis that a visit to the target facilities led to elevation of smoking-related biomarkers.

The adverse health impact of THS has already been scientifically proven [10,11,12], although the specific level of exposure that produces effects in the human body remains unknown. As THS does not involve direct exposure to cigarette smoke, the degree of exposure to THS may be lower than that to SHS. Nevertheless, previous studies have demonstrated that THS exposure occurs in various places of daily life at a level that has an impact on the human body. Living with a smoker [17] and a one-night stay at a hotel [13] resulted in an elevated concentration of smoke-related substances in the bodies of nonsmokers. Matt et al. [23] discovered that a 4 h casino visit can lead to elevated finger nicotine and urine cotinine concentrations because of THS. They also demonstrated that the degree of increase in those concentrations could be lowered with smoke-free policies in casinos. However, the participants in that study stayed overnight at a hotel, which differs from our study model. In our study, participants stayed at the target facilities for only 2 h, during which time their CUC concentrations were elevated. Thus, our results strengthen the evidence for THS exposure after short stays at non-smoke-free facilities.

Despite the adverse health impact of THS, it is challenging to prevent, as there is a perception that exposure to passive smoking only occurs if cigarette smoke can be seen or smelled. However, our study revealed that the smell of smoke does not necessarily correlate with dust NNK concentration, and that there were effects from the interaction between the smell of smoke the increase in CUC concentration. This means that judgement of non-smokers by the smell of cigarettes to avoid exposure to smoke-substances is not an appropriate way to avoid exposure. It should not be assumed that the absence of tobacco odor would result in less exposure to THS. In other words, even when a person does not detect cigarette smoke, residual smoke-related substances in public spaces can cause THS and increase the concentrations of such substances in the body. Therefore, unlike SHS, which is easy to avoid, THS is difficult to avoid and may not even be recognized as something to be avoided. For this reason, there is a need to increase the awareness of the risk of THS, potential exposure routes, and ways to avoid it. In fact, parents who realize that THS is hazardous reportedly ameliorate their smoking behaviors or adopt a smoke-free policy at home [24,25]. Simple educational interventions or social media-based campaigns can increase the public’s knowledge of THS, improve their self-efficacy in terms of THS avoidance, and bring about positive behavioral changes [26,27,28].

Increased social awareness of the risk of THS can drive efforts to strengthen tobacco control policies. Since adopting the FCTC, smoke-free legislations have contributed to the protection of people in many countries from the hazards of SHS. However, such laws primarily regulate public places and do not fully protect nonsmokers from SHS or THS [29]. For example, noraebang—one of the target facilities of this study—are legally recognized smoking zones in Korea. Thus, employees of noraebang are exposed to higher levels of SHS than the general public [19]. On the other hand, frequent users of noraebang use private rooms for individuals or groups (Appendix A illustrates the details pertinent to noraebang in Korea) and, thus, are less likely to be exposed to SHS from other parties. Our participants were not exposed to SHS, and majority did not smell cigarette smoke. Thus, we speculate that participants who visited a noraebang were exposed to THS from smoking by another person in the same room prior to their arrival. On the other hand, internet cafés are legally designated as smoke-free zones in Korea since 2014. However, separate smoking rooms may be designated. Such rooms must be enclosed by walls that prevent smoke from leaking into the computer rooms and must be equipped with a ventilation system that releases the smoke outside the facility (Appendix A illustrates the details pertinent to internet cafés in Korea). Participants who visited internet cafés might have been exposed to smoke-related substances by coming into contact with a smoker who used the smoking room or to the staff managing both smoking and smoke-free zones. Even if the participants could not smell any cigarette smoke, it is possible that the smoke leaked from the smoking room into the smoke-free zones. Our study suggests that smoke-related substances that adhere to the surrounding environment lead to THS exposure among nonsmokers who use the same facility at a different period, and that measures to prevent leakage of smoke are imperfect [30]. Ultimately, the best way to prevent THS is to prohibit smoking in the entire facility, instead of designating smoking and smoke-free zones. Comprehensive tobacco regulation policies need to be implemented to completely ban smoking indoors.

This study had several limitations. First, external factors that cause exposure to smoke-related substances could not be eliminated. Elevated CUC concentrations might have been caused by other factors, such as spending time with smokers or having a lifestyle that increases the exposure to cigarette smoke. We attempted to address this limitation by educating participants about THS and instructing them to do their best to avoid smoke-related substances three days prior to the study. A control group is required to overcome this limitation. Studies should include groups of individuals who do not visit the target facility or include measurement of substances with a longer half-life (e.g., hair nicotine and urine 4-[methylnitrosamino]-1-[3-pyridyl]-1-butanol) and concentrations that do not change within 1–3 days. Another option to enhance the power of evidence is to use an interrupted time-series design, where measurements are obtained several times before and after each visit. Second, our study was conducted in only two facilities in Korea, limiting the generalizability of our results. Tobacco control measures differ among countries, and the degree of THS exposure may differ in the same setting among countries. A short stay in a facility with low concentrations of smoke-related substances may not yield an increase in the concentration of the substances in the body.

## 5. Conclusions

In this study, we evaluated whether a short stay in a facility with smoke-related substances that adhere to the surrounding environment would expose individuals to THS even if they do not smell or are directly exposed to cigarette smoke. We confirmed that even two relatively short stays (approximately 2 h each) in a facility in which people had previously smoked can lead to THS exposure. This was true regardless of the perceptible presence of cigarette smoke.

## Figures and Tables

**Figure 1 toxics-10-00307-f001:**
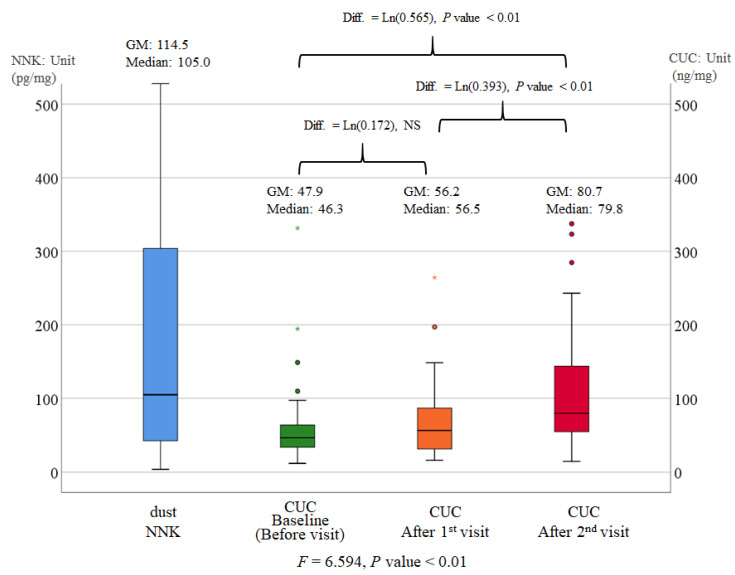
Dust NNK concentration and CUC concentration after the first and second visits. Repeated–measures analyses of variance and post-hoc Fisher’s least significance distance tests were performed. The analysis was performed after values were log-transformed. CUC: creatinine-corrected urine cotinine, GM: geometric mean, NNK: nicotine-derived nitrosamine ketone, and NS: non-significant.

**Figure 2 toxics-10-00307-f002:**
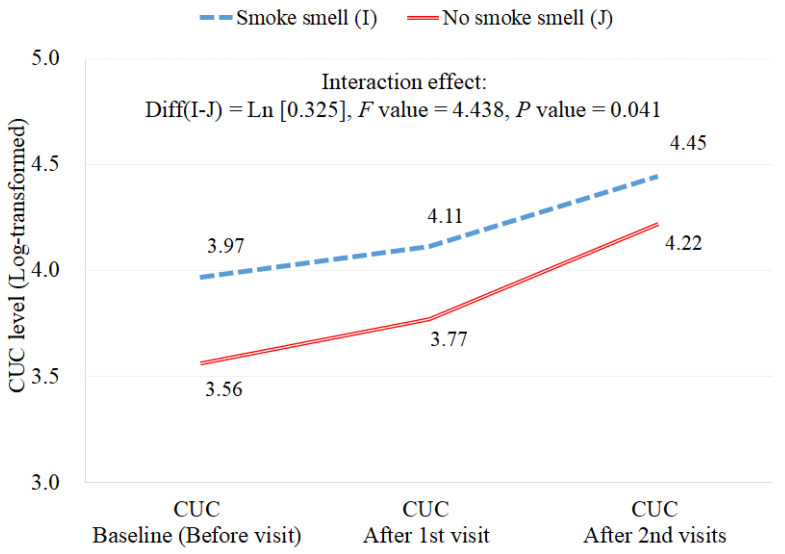
Interaction effect according to the presence or absence of cigarette smell in the facility. The analysis was performed for log-transformed concentrations. CUC: creatinine-corrected urine cotinine.

**Table 1 toxics-10-00307-t001:** Characteristics of participants and target venues and duration of stay.

Characteristics		*n* (%)
Sex	Male	21 (46.7)
	Female	24 (53.3)
Age, mean (95% confidence interval)		22.1 (20–25)
Target venue	Internet café	23 (51.1)
	Noraebang	22 (48.9)
Ventilation condition of the venue	Windows	5 (11.1)
	No windows	40 (88.9)
Smell of smoke upon entering the venue	Yes	11 (24.4)
	No *	34 (75.6)
Duration of stay, minute, mean (95% confidence interval)	1st visit	120.8 (112–140)
	2nd visit	122.1 (120–140)

* Comprises of respondents who answered, “not at all” or “not really” to the question “Did you smell smoke upon entering the venue?”.

**Table 2 toxics-10-00307-t002:** Pearson’s correlation of NNK with CUC concentration changes and the smell of smoke.

		NNK	CUC Concentration Changes *			Smoke Smell ^†^
			0–1	0–2	1–2	
NNK		1				
CUC concentration changes *	0–1	0.144	1			
	0–2	0.562 ^‡^	0.639 ^‡^	1		
	1–2	0.544 ^‡^	−0.289	0.076	1	
Smoke Smell		0.093	−0.157	0.075	0.062	1

* 0–1: difference between CUC concentration after baseline and after the first visit, 0–2: difference between CUC concentration after baseline and after the second visit, and 1–2: difference between CUC concentration after the first and second visits—after 1st visit ^†^ Smoke smell: 0 = none and 1 = smell ^‡^ *p* < 0.001 for log-transformed concentrations. CUC: creatinine-corrected urine cotinine and NNK: nicotine-derived nitrosamine ketone.

## Data Availability

The processed data is available from the author on reasonable request.

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
