# Peer review of "Evaluation of Thirdhand Smoke Exposure after Short Visits to Public Facilities (Noraebang and Internet Cafés): A Prospective Cohort Study"

_toxics, 2022, doi:10.3390/toxics10060307_

Round 1

Reviewer 1 Report

The study from Park and Sim is on an important topic  regarding third hand smoke exposure. However, there are significant queries that leads to questioning the comprehensiveness of the article.

  1. How was the sample size decided?
  2. Median and IQR  values.
  3. The authors mention pre determined internet cafes and Karaoke rooms. Did all participants visit the same? This has to do with exposures and exposure levels.
  4. The data is not seggregated genderwise. age ? 
  5. control group.

Overall it looks to be a pre-study and can be used as a preliminary data. Too many important methodological coordinates are missing.

Author Response

We appreciate the time and effort that you have dedicated to providing your valuable feedback on my manuscript. 
We have been able to incorporate changes to reflect most of the suggestions provided. 
Please see the attachment for a point-by-point response to the comments and concerns.

Reviewer 2 Report

The goal of the present manuscript is to evaluate whether a short term stay in facility with smoke related substances would expose individuals to third hand smoke.  The manuscript addresses a significant question regarding short term exposure to third hand smoke potential exposure in public facility. However, several issues need to be addressed:

  1. It was not clear when the authors measured NNK. It would be beneficial to measure at baseline, first visit, and last visit.
  2. How the authors control for the exposure to SHS. It seems that at baseline the participants has some cotinine in their urine. Is it possible that the increase in CUC could be to other exposure sources? What is the CUC cutoff value to differentiate between THS and SHS? If would be more powerful to add negative controls and measure CUC at three time points without entering these facilities.
  3. The use of other tobacco products such as e-cigarette should be addressed in the participants.
  4. Do the authors notice any differences based on sex (females vs. males)?

Author Response

(The authors gave the same response as above.)

Reviewer 3 Report

thank you for the possibility to review this interesting study.

I really like the idea of this research. In general,  the article is of high practical importance. With such scientific evidence, it would be easier to demand total nicotine prohibition in public places. The article is written in an understandable language. The results are presented in a clear way. 

The only remark is a small group of people studied, which the authors emphasize in the paragraph on limitations of the research.

Line 303-304 - please, complete the reference.

In my opinion this article is worth publishing. 

Author Response

(The authors gave the same response as above.)

Reviewer 4 Report

The residues of smoke-related substances in public places can cause THS and increase the concentration of these substances in the body. Thirdhand smoke is difficult to avoid and may not even be recognized.
The paper is well written and gives a good overview of the issue. 
The presented results are good and clear commented. The originality and novelty of the research are well highlighted.

In my opinion, in the presented form the manuscript  entitled ‘Evaluation of thirdhand smoke exposure after short visits to karaoke rooms and internet cafés: a prospective cohort study’ described by  Myung-Bae Park  and Boram Sim can be recommended for publication in Toxics after minor revision. 
 My remarks and recommendations are as follows:
- In Line 139 "Participant Characteristics" should be "Characteristics of participants and target venues".
- In Table 1, add description to the value 122.1 (120-140) – “2nd visit”
- It is necessary to make more visible signatures/legends on Figure 2.

Author Response

(The authors gave the same response as above.)

Round 2

Reviewer 2 Report

The authors addressed all my comments.